# Quantifying viral pandemic potential from experimental transmission studies

**Elizabeth D. Somsen**[1]*, **Kayla M. Septer**[2], **Cassandra J. Field**[2], **Devanshi R. Patel**[2], **Anice C. Lowen**[3,4], **Troy C. Sutton**[2,4], **Katia Koelle**[4,5]

1 Graduate Program in Population Biology, Ecology, and Evolution, Emory University, Atlanta, Georgia, United States of America, 2 Department of Veterinary and Biomedical Sciences, The Pennsylvania State University, State College, Pennsylvania, United States of America, 3 Department of Microbiology and Immunology, Emory University, Atlanta, Georgia, United States of America, 4 Emory Center of Excellence for Influenza Research and Response (Emory CEIRR), Atlanta, Georgia, United States of America, 5 Department of Biology, Emory University, Atlanta, Georgia, United States of America

* esomsen@emory.edu

## Abstract

In an effort to avert future pandemics, surveillance studies aim to identify animal viruses at high risk of spilling over into humans. These studies have revealed substantial diversity in identified viruses. However, the number of tools currently available to assess pandemic risk is limited. Methods currently in use include the characterization of candidate viruses using *in vitro* laboratory assays and experimental transmission studies in animal models. However, transmission experiments yield relatively low-resolution outcomes that are not immediately translatable to projections of viral dynamics at the level of a host population. To address this gap, we present an analytical framework to extend the use of measurements from experimental transmission studies to generate more quantitative risk assessments. Specifically, we use within-host viral titer data from index and contact animals to estimate parameters relevant to transmission between pairs of individuals. We then extend this model to estimate epidemiological parameters, such as reproduction numbers and generation intervals. We present our analytical framework in the context of two influenza A virus (IAV) ferret transmission experiments: one using influenza A/California/07/2009 (Cal/2009) and one using influenza A/Hong Kong/1/1968 (Hong Kong/1968). In a head-to-head comparison, we find that Cal/2009 has higher pandemic potential than Hong Kong/1968. Our results depend on several assumptions, including that within-host viral dynamics in humans and those in the model animal used (here, ferrets) share quantitative similarities and that viral transmissibility between model animals reflects viral transmissibility between humans. The methods we present to assess pandemic risk of viral isolates can be used to improve relative risk assessment of other emerging viruses of pandemic concern.

**Data availability statement:** All code written in support of this publication is publicly available at https://github.com/esomsen/quantifying_pandemic_potential_code.

**Funding:** This work was supported by the National Institute of Allergy and Infectious Diseases of the National Institutes of Health under the award numbers F31AI186550 (E.D.S.) and by the National Institute of Allergy and Infectious Diseases Centers of Excellence for Influenza Research and Response (CEIRR) contract 75N93021C00017 (K.K., T.C.S., A.C.L.). Research reported in this publication was further supported by Emory University and the Infectious Diseases Across Scales Training Program T32AI138952 (E.D.S.). These studies were also supported by the USDA National Institute of Food and Agriculture, Hatch project 4955 (T.C.S.). The content is solely the responsibility of the authors and does not necessarily represent the official views of Infectious Diseases Across Scales Training Program, Emory University, The Pennsylvania State University, the National Institutes of Health, or the USDA. The funders had no role in study design, data collection and analysis, decision to publish, or preparation of the manuscript.

**Competing interests:** The authors have declared that no competing interests exist.

## Author summary

Pandemic viruses often originate in animal reservoirs. Surveillance studies at the animal-human interface aim to identify viruses in animals that may be of potential concern in sparking a pandemic among humans. Experimental transmission studies in animal models can then be used to further gauge the pandemic risk of the viruses identified in these surveillance studies. However, only simple outcomes usually get reported from these experiments, such as the fraction of exposed animals that become infected. Here, we develop a framework to extract more information from these experiments and apply this framework to two experimental influenza virus transmission studies in ferrets. We then use our results to estimate epidemiological parameters commonly used in public health to quantify pandemic potential. Our approach allows us to use commonly gathered data from experimental transmission studies to compare the pandemic risk of animal viruses before they emerge in humans. It further allows us to predict characteristics of epidemic dynamics in humans in the event of an emerging pandemic.

## Introduction

Pandemics arising via zoonotic spillover from animal reservoirs have resulted in mass human mortality and extreme societal disruption. Notable examples are the 1918 Spanish flu pandemic caused by influenza A virus (IAV) H1N1 and the COVID-19 pandemic caused by SARS-CoV-2. Given the impact of pandemics, there is considerable interest in preventing pathogen emergence, particularly of rapidly spreading respiratory pathogens. One key component in preventing pathogen emergence involves surveillance of reservoir species to identify viruses with pandemic potential. However, reservoir species can harbor a large diversity of viral pathogens, resulting in a substantial pool of possible pre-pandemic strains. Approaches to winnow down the considerable diversity of candidate viruses to those that are more likely to cause a pandemic are therefore essential in guiding control efforts to mitigate pandemic risk. One approach for prioritizing viruses involves genotypic and phenotypic characterization of viral strains to identify those with traits associated with pandemic risk [1,2]. For example, assays measuring IAV hemagglutinin receptor binding specificity are used to determine if an influenza virus preferentially binds cells with $\alpha$-2,3 or $\alpha$-2,6 sialic acid linkages. IAVs with $\alpha$-2,3 binding preference generally do not transmit well from person to person. For many relevant traits, however, genotypic characterizations provide only limited information because the relationship between viral genotypes and their phenotypes has not been fully characterized. Risk assessments based on genetic analyses are therefore limited to traits where the genetic basis is well understood.

Experimental transmission studies have also been widely used to estimate the transmission potential of viral strains. For IAV risk assessment, ferrets are often used because of similarities between ferrets and humans in phenotypes relevant to infection and transmission, such as viral attachment [3] and clinical signs. A notable study found evidence for a strong positive correlation between ferret-to-ferret transmission

probability (as measured by secondary attack rate) and IAV circulation patterns in humans [4]. The findings of this study were particularly impactful because numerous ferret experiments were used in the assessment of the relationship between secondary attack rates and IAV circulation patterns in humans. Results from these types of transmission experiments provide semi-quantitative measurements of transmission potential, one aspect of pandemic risk. However, the read-outs of these experiments are typically low-resolution estimates. For example, in a study with three index-contact pairs, the observed overall transmission efficiency could be 0% (0/3), 33% (1/3), 67% (2/3), or 100% (3/3). These small sample sizes can hamper statistical comparisons of transmission efficiencies between different viral isolates. For example, Nishiura and colleagues have shown that demonstrating significant differences in transmission potential between two experimental groups requires at least four transmission pairs per group [5]. Moreover, a significant difference with four pairs per group can only be found when one experiment has full transmission (4/4 contact animals become infected) and the other has no transmission (0/4 contact animals become infected).

Given the cost and effort involved in carrying out these experimental transmission studies, it would be worthwhile to extract more information about transmission from the data that are commonly collected during these studies. To this end, we here develop an analytical framework to improve risk assessment using within-host viral titer measurements. Our approach relies on the titer data from infected index animals as well as infected contact animals to provide quantitative estimates of transmission parameters and epidemiological components of pandemic risk. To motivate and illustrate our approach, we apply it to data from two recently performed IAV experimental transmission studies [6], the first using a 2009 pandemic H1N1 isolate and the second using a 1968 pandemic H3N2 isolate. Below, we first present the data from these two experimental transmission studies and then develop our framework in application to these data.

## Results

### Observed infection dynamics in the experimental IAV transmission studies

The transmission studies analyzed here have previously been described in detail, and their data have been made publicly available [6]. An overview of the study design is provided in the Methods section. In brief, index ferrets were inoculated with influenza A virus at various doses, ranging from $10^0$ to $10^6$ $TCID_{50}$. The experimental design allowed only airborne transmission of the virus from an index animal to a paired contact animal. We considered an animal to be infected if viral shedding over the limit of detection ($10^1$ $TCID_{50}$/mL) was observed at any sampled timepoint. Transmission efficiency was defined as the fraction of contact ferrets that were infected by their paired index in the subset of pairs in which the index became infected.

Transmission efficiencies ranged between 75-100% for influenza A/California/07/2009 (Cal/2009) and between 25-100% for influenza A/Hong Kong/1/1968 (Hong Kong/1968) (Fig 1). Interestingly, the transmission efficiencies for Cal/2009 fell within the range of respiratory droplet attack rates that Buhnerkempe and colleagues [4] would predict as having a high probability of a supercritical classification in humans. In contrast, the efficiencies observed in the Hong Kong/1968 transmission study spanned a range between subcritical and supercritical classification in humans.

### Estimation of the force-of-infection function

Several qualitative patterns emerge from inspection of the viral titer dynamics shown in Fig 1. First, the probability of transmission to a contact animal generally appears to be lower when viral titers in the index are lower. Second, particularly in the Cal/2009 study, transmission to a contact animal appears to occur earlier at higher inoculum doses. This is likely because higher inoculum doses tend to result in higher viral titers in the index animals shortly following challenge. As such, viral titer levels appear to impact not only *whether* a contact animal gets infected but, in the case of infection, *when* it gets infected. This observation is key to the development of our analytical approach.

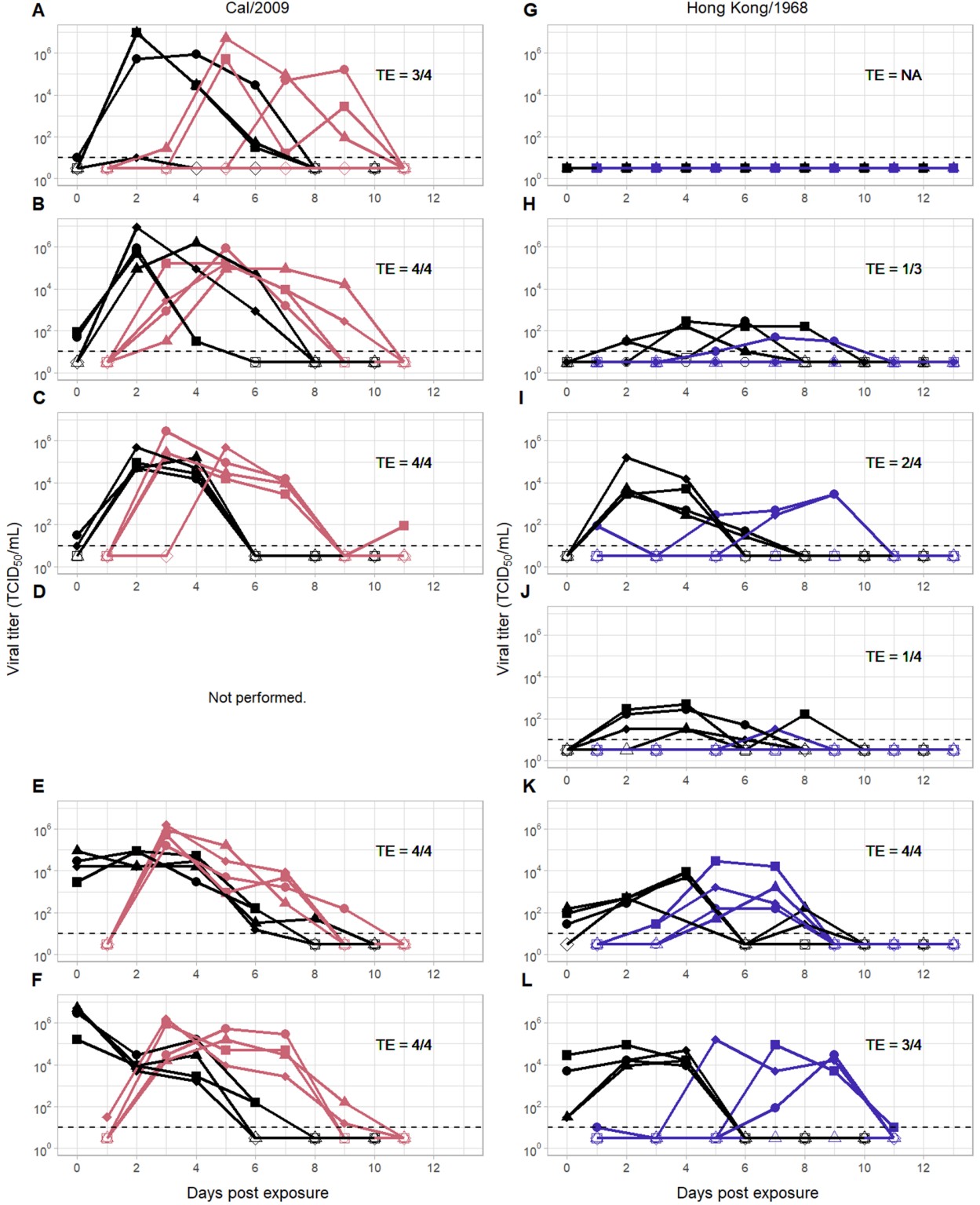

**Fig 1. Viral kinetics in the Cal/2009 and Hong Kong/1968 experimental transmission studies.** The left column shows Cal/2009 viral kinetics in index ferrets (black) and contact ferrets (pink). The right column shows Hong Kong/1968 viral kinetics in index ferrets (black) and contact ferrets (blue). Rows correspond to different index inoculum doses: (A, G) $10^0$, (B, H) $10^1$, (C, I) $10^2$, (D, J) $10^3$, (E, K) $10^4$, and (F, L) $10^6$ TCID$_{50}$. The limit of detection

is shown as a dotted line at $10^1$ TCID$_{50}$/mL. Transmission pairs share the same marker. Open markers at $10^{0.5}$ TCID$_{50}$/mL denote samples with viral titers that fall below the limit of detection. Each panel further provides the transmission efficiency (TE) in that experiment, with the denominator denoting the number of infected index animals in the experiment and the numerator denoting the number of corresponding contact animals that became infected. Data reproduced from [6].

Given that the viral kinetics of index animals appear to impact both the probability and the timing of transmission to paired contacts, we here develop a framework for quantifying the impact of index viral kinetics on transmission to contacts. A previous study has considered a constant force-of-infection model to quantify transmission rates between index and contact animals in experimental transmission studies [7]. Here, we instead use a time-varying force of infection to evaluate how transmission between index and contact depends on time-varying index viral titers. We define the force of infection $\lambda$ that an index animal exerts on a contact animal at time $t$ as some functional form that depends on the index animal's viral titer at time $t$. For example, we can let $\lambda$ linearly depend on the $log_{10}$ of the index animal's viral titer. (This functional form is empirically supported by some previous studies [8–10].) In this case, the force of infection is given by:

$$\lambda(t) = \begin{cases} 0 & \text{if } V(t) < LOD \\ s \times log_{10}(V(t)) & \text{if } V(t) \geq LOD \end{cases} \tag{1}$$

where $V(t)$ denotes the index animal's viral load measured at time $t$ in units of TCID$_{50}$/mL, and LOD denotes the limit of detection of the assay used to quantify viral titers. The parameter $s$ quantifies a component of transmissibility: the higher the value of $s$, the larger the force of infection from the index animal at time $t$. While $s$ contributes to the force of infection experienced by a contact animal, the force of infection also depends on the index animal's viral titer. The probability that an index transmits to its contact and the time at which the contact becomes infected are both functions of this time-varying force of infection (Methods). Given observed index viral titers (Fig 1) and information on when a contact initially becomes infected, we can statistically estimate the parameter $s$ (Methods). Fig 2A shows the estimates of $s$ derived from the Cal/2009 and Hong Kong/1968 transmission studies. For Cal/2009, the maximum likelihood estimate is $s = 0.111$ (95% confidence interval (CI) = $[0.068, 0.172]$) and for Hong Kong/1968, it is $s = 0.047$ ($[0.024, 0.082]$). To determine whether $s$ of Cal/2009 is statistically larger than that of Hong Kong/1968, we additionally estimated a single value for $s$ by combining the two datasets. We then calculated the support for this simpler, one-parameter model relative to the two-parameter model with different $s$ estimates (one for Cal/2009 and one for Hong Kong/1968) (Methods). We found that the model with two $s$ estimates was statistically preferred, but the extent to which it was preferred was minor ($\Delta AICc = 2.74$).

To gain better intuition into the extent to which the inferred forces of infection of Cal/2009 and Hong Kong/1968 differed from one another, we calculated the probability of a contact becoming infected if exposed for a period of one day to an index animal with a given viral titer (Fig 2B). Given the values of $s$ for Cal/2009 and for Hong Kong/1968, the probability that a contact becomes infected during this day of exposure is predicted to be higher (approximately twice as high) for Cal/2009 than for Hong Kong/1968 at a given index viral titer.

## Assessment of alternative force-of-infection functional forms

Because we do not *a priori* know that the $log_{10}$ functional form for the force of infection that we used in Equation 1 is an appropriate one to use, we next considered alternative functional forms for the relationship between force of infection $\lambda$ and viral titers. The alternative functional forms we considered were a basic linear relationship, a threshold relationship (whereby $\lambda$ was zero below a threshold viral titer and some constant value above that threshold titer), and a flexible Hill function (Methods). We estimated the parameters for each of these models for each viral isolate. Fig 3 shows these force

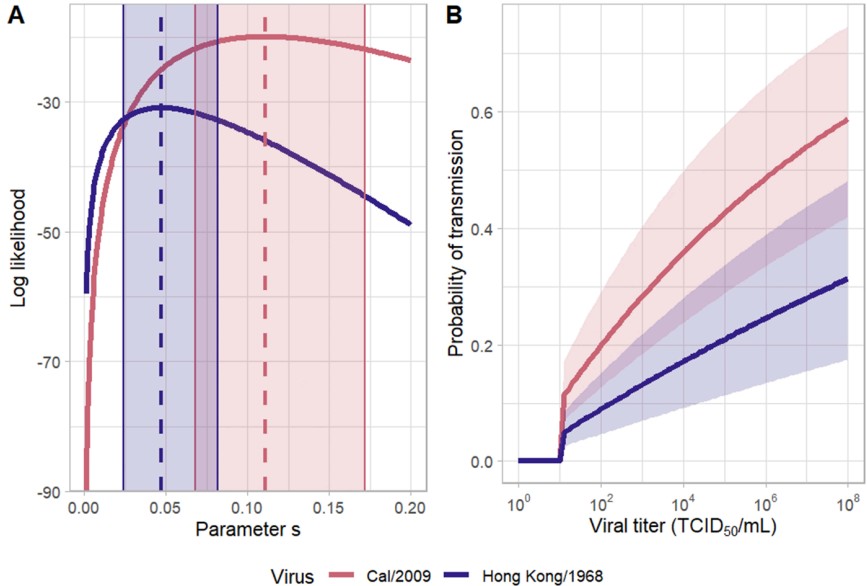

**Fig 2. Statistical estimation of the $log_{10}$ force-of-infection function.** (A) The log likelihood profile for parameter $s$ for both the Cal/2009 virus (pink) and the Hong Kong/1968 virus (blue). Dashed lines show the maximum likelihood values of $s$. Shaded regions show the 95% confidence intervals. (B) Calculated probabilities of transmission to a contact after a one-day exposure to an index animal with a constant viral titer, as given on the x-axis. Solid lines show the probabilities calculated using the maximum likelihood estimates for $s$. Shaded regions show transmission probabilities that fall within the 95% confidence interval values for $s$.

of infection estimates for the $log_{10}$ model and the alternative models for each of the two viral isolates, along with calculated probabilities of transmission given a one-day exposure period to an index animal with a given viral titer. To determine which of these models performed best, we then calculated the AICc scores for each model (Table 1). The linear form performed very poorly for both Cal/2009 and Hong Kong/1968. The threshold model had the best AICc score for both isolates, but there is little statistical support for discriminating between the $log_{10}$, threshold, and Hill forms. Consistent with this, the force of infection and probability of transmission results shown in Fig 3 for these three forms demonstrates that their results are quite similar.

## Quantification of parameters at the transmission event

Whereas the index animals in the Cal/2009 and Hong Kong/1968 experimental transmission studies received direct intranasal challenge with varying inoculum doses, the contact animals acquired their infections via inhalation of infectious respiratory particles that were shed by the index animals. As such, one could argue that the viral titer dynamics observed in the contact animals are more likely to resemble natural infections. Indeed, a qualitative re-inspection of Fig 1 indicates that, for Cal/2009, neither the peak viral titer nor the duration of infection in contact animals appears to depend on the index's inoculum dose. This finding is also statistically supported, with a lack of positive correlation between index inoculum dose and peak viral titer ($p = 0.62$; Fig B in S1 Text) and a lack of positive correlation between index inoculum dose and overall duration of infection ($p = 0.49$ and $p = 0.30$; Fig B in S1 Text). In the Hong Kong/1968 study, there appears to be considerably more heterogeneity across the contact animals in their viral titer dynamics. However, we do observe a significant positive correlation between index inoculum dose and peak viral titers ($p = 0.01$; Fig B in S1 Text). We also find a weak positive correlation between index inoculum dose and overall duration of infection ($p = 0.04$; Fig B in S1 Text), but this correlation is lost when we re-calculate the duration of infection for two contact animals that have transiently positive viral titers ($p = 0.68$; Figs A-B in S1 Text). As such, there is little overall statistical support for the inoculum doses of

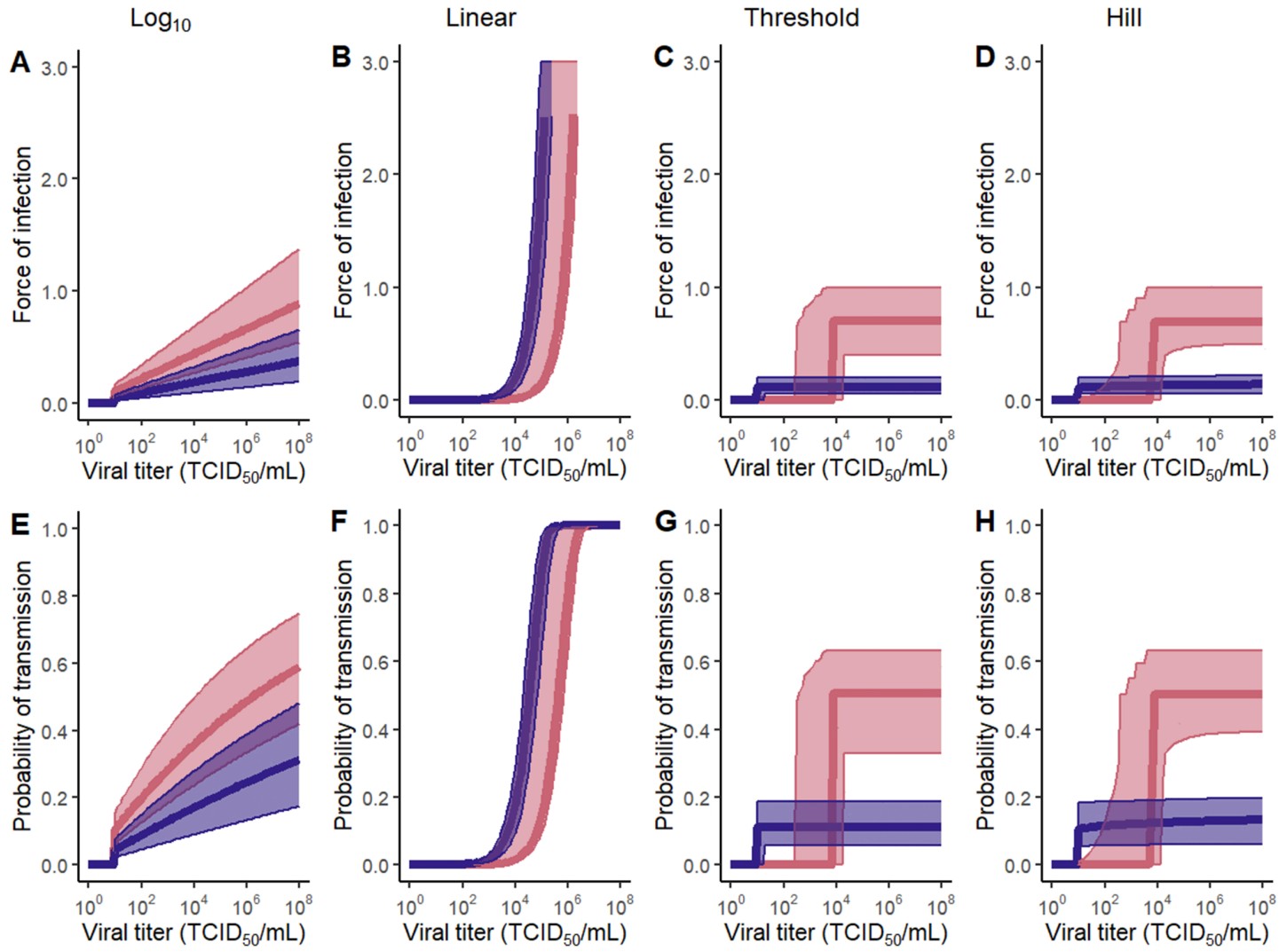

**Fig 3. Statistical estimation of different force-of-infection functional forms.** (A)-(D) Estimated forces of infection ($\lambda$ values) across a range of viral titers. (E)-(H) Probabilities of transmission given a one-day exposure to a constant viral titer, given the $\lambda$ estimates from (A)-(D), respectively. Columns correspond to alternative functional forms: (A,E) $log_{10}$-transformed, (B,F) linear, (C,G) threshold, and (D,H) Hill function. Solid lines show results using maximum likelihood parameter estimates given in Table A in S1 Text. Shaded regions show 95% confidence intervals.

**Table 1. Model comparison of original $log_{10}$ and alternative force-of-infection functional forms.** The columns specify the functional form, the model parameters, the corrected AIC (AICc) value for the model fit to the Cal/2009 data, and the AICc value for the model fit to the Hong Kong/1968 data. Models with lower AICc are preferred. ΔAICc values are also shown in parentheses for both viral isolates, relative to the statistically most preferred model, whose AICc values are shown in boldface.

| Functional Form | Parameters | Cal/2009 AICc | Hong Kong/1968 AICc |
|---|---|---|---|
| $log_{10}$ | $s$ | 42.22 (3.77) | 64.12 (0.99) |
| linear | $s_L$ | 95.2 (56.75) | 122.59 (59.46) |
| threshold | $h, r$ | **38.45** | **63.13** |
| Hill | $q, k_a, n$ | 41.54 (3.09) | 66.05 (2.92) |

the index animals, or their resultant viral titer dynamics, impacting features of the contact animal viral titers beyond times of infection. In both studies, viral titers in the contact animals appear to start off at low levels, reaching their peaks within 2 to 4 days. Here, to quantify parameters at the transmission event, we therefore assume that the viral titer dynamics observed in the contact animals are representative of those in natural infections. Moreover, for projecting relative transmission potential, we need to assume that the contact animal viral dynamics are not only similar to those of natural ferret infections with the isolates considered, but also similar to those of natural human infections with the isolates considered. This is not a trivial assumption, and we return to this point in the Discussion.

Our approach to projecting parameters across the transmission event relies on combining viral titer dynamics observed across contact ferrets with our parameterized force-of-infection function. These contact ferrets become "theoretical" index animals, and we use their observed titer dynamics and the force-of-infection function to simulate the expected number of secondary infections (Methods). Fig 4A and 4B show a single stochastic realization of the offspring distribution for Cal/2009 and Hong Kong/1968 "theoretical" index animals, respectively, assuming an average of 15 hour-long contacts per day and using the $log_{10}$ functional form, parameterized with $s$ values randomly drawn from the estimated 95% confidence interval of this parameter. We chose a rate of 15 contacts per day based on contact rate estimates for children under the age of 18 [11]. We chose the $log_{10}$ force-of-infection function due to its support from previous studies and its good performance among the alternative models considered (Table 1). By eye, both offspring distributions appear to follow a negative binomial distribution with different means. From our stochastic simulations, we could also keep track of when transmissions occurred from each infected animal and use this information to calculate the generation interval, defined as the time between the infected animal's infection and onward transmission. Fig 4C and 4D show these generation times for the Cal/2009 and Hong Kong/1968 secondary infections that were realized and shown in Fig 4A and 4B, respectively.

To scale up from the single stochastic realizations shown in Fig 4A–4D for the populations of Cal/2009 and Hong Kong/1968 "theoretical" index ferrets, we performed 1000 stochastic realizations for each virus. To ensure that our simulations incorporated uncertainty in the value of $s$, we sampled values of $s$ from the likelihood profile we identified previously (Fig 2A) using a Metropolis-Hastings algorithm (Methods). Each of the 1000 simulations included all of the contact animals that were infected in the original study ($n$=19 for Cal/2009 and $n$=11 for Hong Kong/1968). For each realization, we fit a negative binomial distribution to the resultant offspring distribution and calculated the mean and the overdispersion parameter $k$ of this distribution. The mean of this distribution is, by definition, equivalent to the reproduction number $R_0$, estimated from a single stochastic realization. Fig 4E plots the distribution of $R_0$ values that were estimated for the Cal/2009 and Hong Kong/1968 virus isolates using the 1000 stochastic realizations. From this figure, it is clear that the $R_0$ estimated for the Cal/2009 virus is substantially higher than the one estimated for the Hong Kong/1968 virus, even with considerable uncertainty in its exact value (Welch's two sample t-test, $p < 0.001$). Moreover, at the contact rate assumed, the $R_0$ estimated for Cal/2009 exceeds one in almost all of the simulations, whereas it exceeds one for Hong Kong/1968 in fewer than half the simulations. This indicates that, at this contact rate, Cal/2009 would have considerable pandemic potential whereas Hong Kong/1968 would not.

Fig 4F shows the cumulative distributions of the overdispersion parameters estimated for Cal/2009 and Hong Kong/1968 from their respective stochastic realizations. For Cal/2009, we find that 2 of the 1000 stochastic simulations have an estimated overdispersion parameter $k$ of less than one. For Hong Kong/1968, around 10% of simulation have an estimated $k$ of less than one. This indicates that there is relatively little transmission heterogeneity for both viral isolates, with transmission heterogeneity being particularly low for Cal/2009. Note that, for both viral isolates, the estimated overdispersion values only capture transmission heterogeneity stemming from interindividual variation in viral titers. As such, our estimates of the extent of transmission heterogeneity will be lower-bound estimates. This is because of additional variation that contributes to transmission heterogeneity, such as heterogeneity in contact rates that occur in real world settings [12].

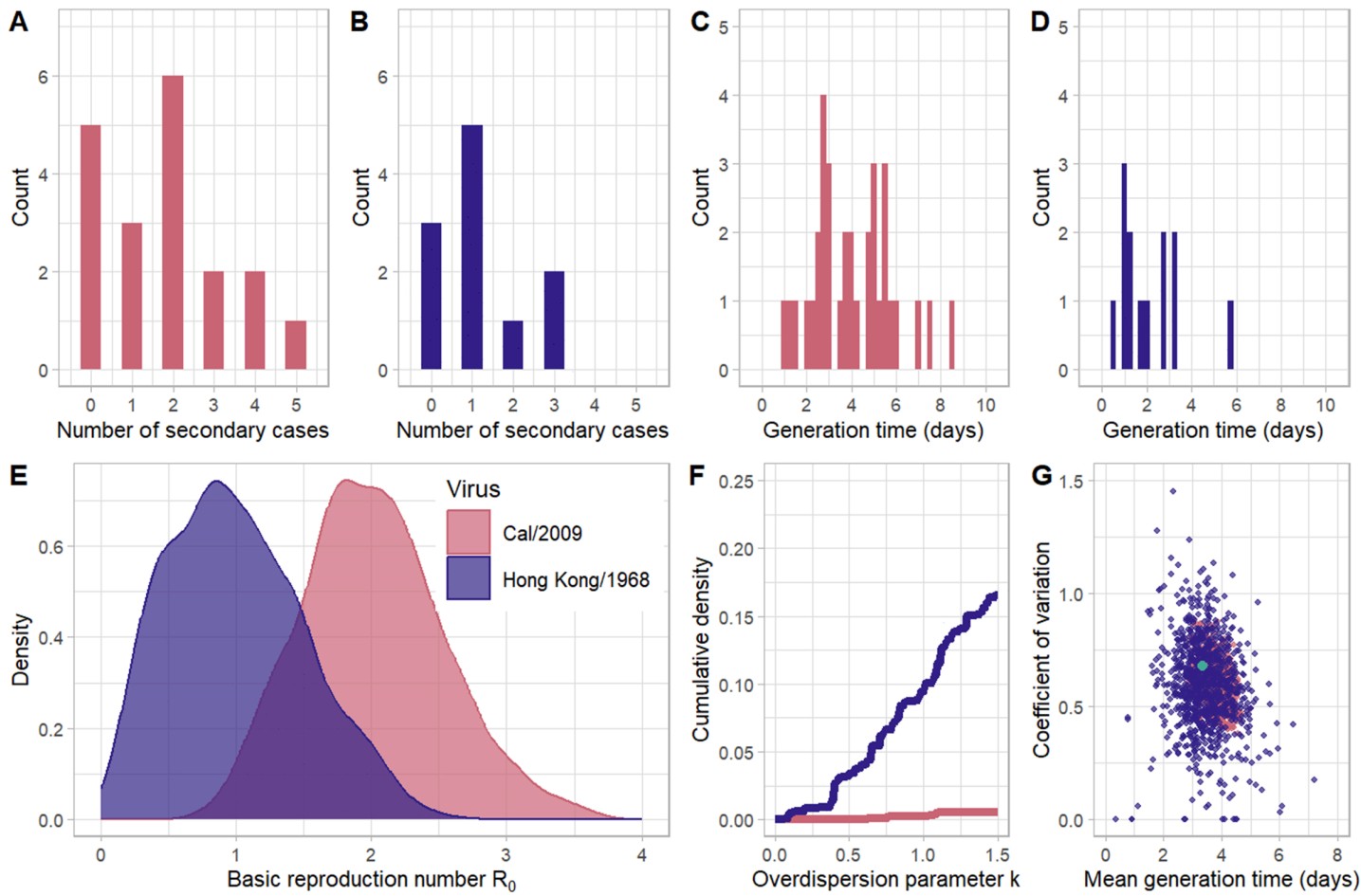

**Fig 4. Estimation of transmission event parameters for Cal/2009 and Hong Kong/1968.** (A) Distribution of the number of secondary infections generated by the $n=19$ infected Cal/2009 contact animals. (B) Distribution of the number of secondary infections generated by the $n=11$ infected Hong Kong/1968 contact animals. In panels (A) and (B), the distributions each show the outcome of one stochastic realization, assuming each of the infected animals has 15 one-hour long contacts per day. (C) The empirical generation interval distribution for the $n=34$ secondary cases generated in the Cal/2009 simulation shown in panel (A). (D) The empirical generation interval distribution for the $n=13$ secondary cases generated in the Hong Kong/1968 simulation shown in panel (B). (E) The distribution of $R_0$ values from 1000 Cal/2009 and 1000 Hong Kong/1968 stochastic simulations. (F) The cumulative distribution of overdispersion ($k$) values from the Cal/2009 and Hong Kong/1968 stochastic simulations that resulted in at least one individual transmitting infection. This was 1000 simulations for Cal/2009 and 997 simulations for Hong Kong/1968. (G) Estimated means and coefficients of variation of the gamma distributions fit to the empirical generation time distributions of each of the 1000 stochastic simulations for Cal/2009 and each of the 997 stochastic simulations for Hong Kong/1968. The teal dot indicates the mean and coefficient of variation used to parameterize the gamma distribution that was used to project the intrinsic growth rates in Fig 5D.

Finally, we fit a gamma distribution to each stochastic simulation, for both Cal/2009 and Hong Kong/1968, and calculated the mean and coefficient of variation from each estimated gamma distribution. These means and coefficients of variation are shown in Fig 4G. For Cal/2009, the median generation interval is 3.77 days ($25^{th}$ percentile = 3.56, $75^{th}$ percentile = 3.97) and the median coefficient of variation is 0.60 ($25^{th}$ percentile = 0.54, $75^{th}$ percentile = 0.66). For Hong Kong/1968, the median generation interval is 3.41 days (3.00, 3.87) and the median coefficient of variation is 0.60 (0.48, 0.71).

In sum, for both of the viral isolates considered, we calculated three epidemiological parameters that are relevant to pandemic risk: the basic reproduction number (under an assumption of a contact rate of 15 individuals per day), the

overdispersion parameter, and characteristics of the generation interval. Generally, we expect that pandemic risk is greater for virus isolates with higher $R_0$, low overdispersion, and short generation intervals (due to more rapid spread). As such, a head-to-head comparison of Cal/2009 against Hong Kong/1968 would indicate that Cal/2009 would pose greater pandemic risk than Hong Kong/1968, largely due to Cal/2009's higher reproduction number. Calculations of these same three parameters for the other three force-of-infection functional forms (linear, threshold, and Hill) are shown in Figs D-F in S1 Text. The results for the threshold and Hill functional forms are largely consistent with those of the $log_{10}$ force-of-infection functional form. The results for the linear functional form show less of a difference between $R_0$ estimated for Cal/2009 and Hong/Kong/1968, as well as much lower estimates of the overdispersion parameter (indicative of more superspreading) and shorter generation times. However, our earlier model comparison (Table 1) indicated that the linear functional form performed substantially worse than any of the three other functional forms considered, and as such we do not put weight on these latter linear functional form results.

### Prediction of pandemic potential and dynamics

Above, we assumed a single contact rate (an average of 15 hour-long contacts per day) to project the number of secondary infections for each of the two viral isolates and to estimate their generation intervals using our parameterized $log_{10}$ force-of-infection function. The chosen contact rate impacts the distribution of secondary infections and thus our estimates of $R_0$. In Fig 5A, we show how our $R_0$ estimates change for Cal/2009 and Hong Kong/1968 across a broad range of contact rates. The range of contact rates we consider span those deemed relevant for a respiratory virus of humans [11].

We can use the $R_0$ estimates shown in Fig 5A to predict the risk of pandemic establishment of the two viruses, given an assumed contact rate (Methods). Fig 5B shows this establishment risk for each virus, under two different transmission heterogeneity assumptions: one with no transmission heterogeneity ($k = \infty$) and one with a moderate level of transmission heterogeneity ($k = 1$) that exceeds our simulated expectations (Fig 4F). For Cal/2009, at contact rates below 8 (hour-long) contacts per day, the virus is not predicted to establish due to a subcritical basic reproduction number ($R_0 < 1$). For Hong Kong/1968, the first contact rate at which $R_0$ exceeds 1 is 16 (hour-long) contacts per day. Increases in daily contact rates above these critical values result in higher probabilities of viral establishment, although the exact probabilities depend on the extent of transmission heterogeneity. As theoretically expected [13], higher levels of transmission heterogeneity lower the probability of viral establishment. In the absence of transmission heterogeneity ($k = \infty$), the probability that Cal/2009 establishes is approximately 89% when the contact rate is 18 contacts per day, corresponding to the contact rate of the age group with the highest number of daily contacts in the study by Mossong and colleagues [11]. At a contact rate of 7 contacts per day, corresponding to the contact rate of the age group with the lowest number of daily contacts, the probability that Cal/2009 establishes is 0%. Likewise, for Hong Kong/1968, the probability of establishment is 26% at 18 contacts per day and 0% at 7 contacts per day.

A previous study has pointed out that initial $R_0$ values for a spillover virus matter for projecting pandemic risk, even if they fall below one [14]. This is because viruses have the potential to adapt as they transmit between individuals along stuttering chains. At $R_0$ values closer to one, the length of these stuttering chains is expected to be longer than at $R_0$ values closer to zero, providing viruses with higher $R_0$ greater opportunity to adapt and thus to ultimately establish. In Fig 5C, we plot for each virus considered, the average length of stuttering chains across the assumed range of contact rates (Methods). We plot these average lengths only for the subset of contact rates for which a virus still has a subcritical ($< 1$) $R_0$. At higher contact rates, the average length of stuttering chains is higher.

Finally, we can use our estimates of $R_0$, shown in Fig 5A, along with a specified generation time distribution, to estimate the intrinsic growth rates ($r$ values) of the viruses under various contact rate scenarios. The intrinsic growth rate of a virus is important to project in that it specifies the viral doubling time at the level of the host population and therefore the speed at which a pandemic takes off [15]. Fig 5D shows intrinsic growth rate estimates for both viruses across the range

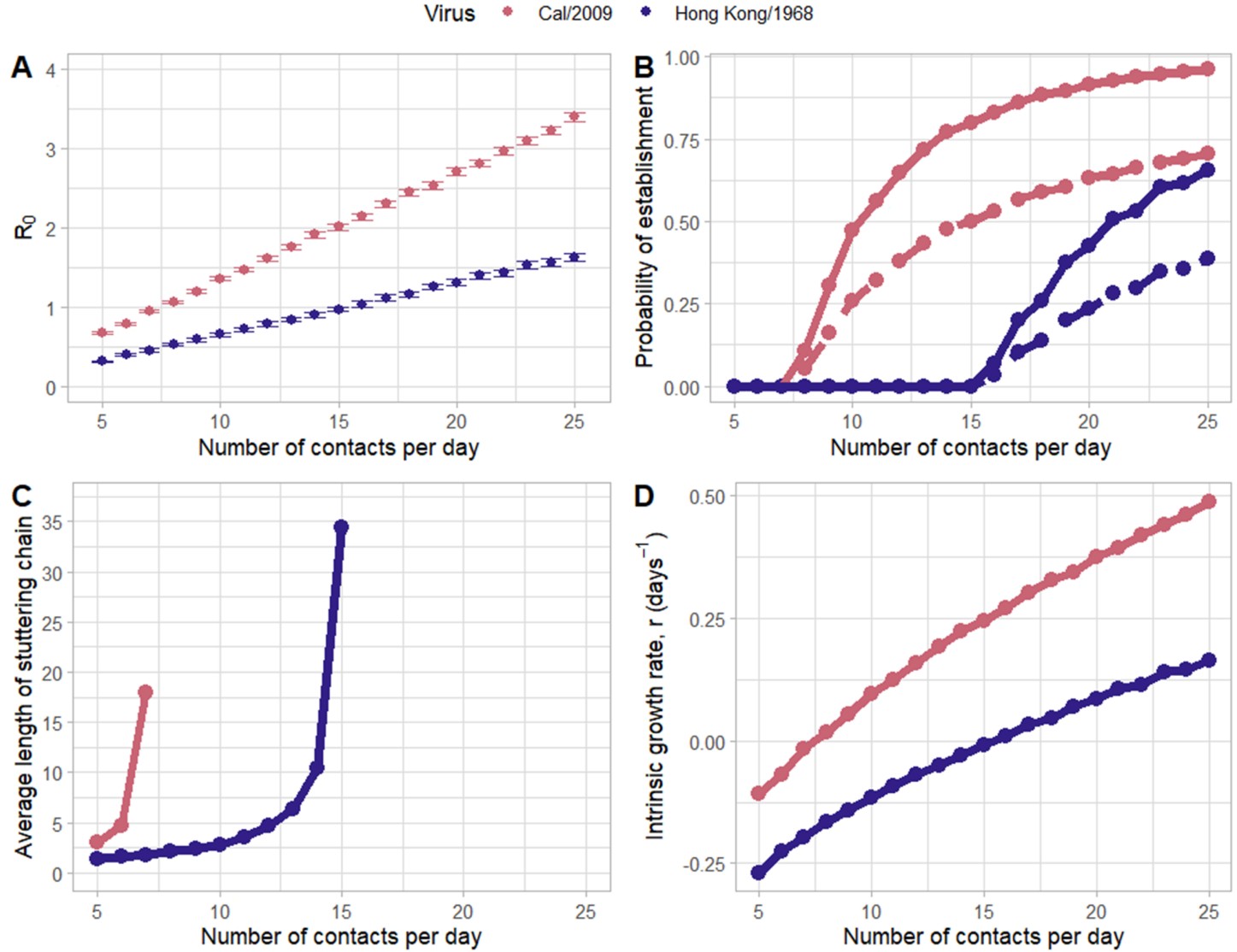

**Fig 5. Prediction of pandemic potential and dynamics.** (A) Estimates of the reproduction number $R_0$ for Cal/2009 and Hong Kong/1968 across a range of contact rates. Error bars indicate 95% confidence intervals. (B) The probability of pandemic establishment for Cal/2009 and Hong Kong/1968 across a range of contact rates. The solid lines indicate probabilities in the absence of transmission heterogeneity ($k = \infty$). The dashed lines indicate probabilities with a moderate level of transmission heterogeneity ($k = 1$). (C) The average length of stuttering transmission chains across the range of contact rates. Average lengths of chains are only shown for contact rates that yield $R_0$ estimates of less than one. (D) Estimated intrinsic growth rates for Cal/2009 and Hong Kong/1968 across a range of contact rates, assuming their generation intervals are gamma distributed with mean = 3.34 days and a coefficient of variation of 0.68 (teal point in Fig 4G). The doubling time of an epidemic decreases as $r$ increases. When $r = 0.07$ days$^{-1}$, the doubling time is 10 days; when $r = 0.14$ days$^{-1}$, the doubling time is 5 days; and when $r = 0.35$ days$^{-1}$, the doubling time is 2 days. In all panels, Cal/2009 projections are shown in pink and Hong Kong/1968 projections are shown in blue.

of assumed contact rates, under the assumption of a mean generation time of 3.34 days and a coefficient of variation of 0.68. At any given contact rate, Cal/2009 is expected to have a shorter doubling time than Hong Kong/1968. For example, at a contact rate of 20 (hour-long) contacts per day, Cal/2009 is projected to have a doubling time of approximately 1.9 days, whereas Hong Kong/1968 has a substantially longer doubling time of approximately 8.4 days. For both viruses, projected intrinsic growth rates are higher at higher contact rates, as expected.

In sum, for both of the viral isolates considered, we quantified several metrics relevant to pandemic risk across a range of contact rates: the probability of epidemic establishment, the average length of stuttering transmission chains, and the intrinsic growth rate of an epidemic. Generally, we expect that virus isolates with $R_0$ values exceeding one across a broad range of reasonable contact rates, those with high probabilities of establishment, those with long stuttering transmission chains, and those with higher intrinsic growth rates all have an elevated pandemic risk. A head-to-head comparison of Cal/2009 against Hong Kong/1968 indicates that Cal/2009 would pose greater pandemic risk than Hong Kong/1968. This is due to the higher $R_0$ that is estimated for Cal/2009 relative to that of Hong Kong/1968, at any given contact rate. This higher projected $R_0$, in turn, is in part a result of the higher viral titers in Cal/2009-infected (contact) animals relative to those of Hong Kong/1968-infected (contact) animals. The higher projected $R_0$ for Cal/2009 is also partly due to the higher transmissibility of Cal/2009, even when controlling for viral titers. This higher transmissibility is reflected in the higher $s$ estimates for Cal/2009 relative to Hong Kong/1968 (Fig 2A).

As in the previous section, we performed analogous projections for the other three force-of-infection functional forms (linear, threshold, and Hill), with results shown in Figs G-I in S1 Text. The results from the threshold and Hill functional forms are qualitatively consistent with those of the $log_{10}$ force-of-infection functional form in that Cal/2009 is projected to have higher pandemic risk than Hong Kong/1968. Because the linear functional form estimates low $R_0$ values for both viruses across the range of contact rates considered (Fig G in S1 Text), projected probabilities of establishment of both viruses are very low and intrinsic growth rates are largely negative. Again, because our earlier model comparison (Table 1) indicated that the linear functional form performs substantially worse than all of the other functional forms considered, we do not put weight on these latter linear functional form results.

## Discussion

Here, we developed an analytical approach to gauge the pandemic potential of zoonotic viruses using viral titer data from experimental transmission studies. Our approach relied on estimating the parameters of a force-of-infection function using information on index animal viral kinetics and the timing of detected contact animal infection. It further relied on assuming that the viral kinetics of contact animals reflected those of animals that would be naturally infected with the viruses considered, and more stringently, those of humans who would become naturally infected with the zoonotic viruses considered. We applied our approach to existing transmission study datasets that used two IAV isolates and found that Cal/2009 had higher pandemic potential than Hong Kong/1968 (Figs 4 and 5). We imagine the primary use case for this approach would be in assessing the relative pandemic potential of animal-origin influenza viruses that have not yet emerged in humans. These results, and indeed the results of any subsequent analyses using this method, are valid only for the specific isolates used in the experimental studies. The continual and rapid evolution of influenza viruses precludes us from generalizing these results to other H1N1 or H3N2 strains, including those which will arise in the future and could potentially cause pandemics.

The analytical approach we presented here affords several benefits. In addition to being able to assess relative pandemic potential of different viral strains, our approach provides insight into why one virus might have higher transmission potential than another. For example, we found that our maximum likelihood estimate of the parameter $s$ was higher for Cal/2009 than it was for Hong Kong/1968 (Fig 2). An individual infected with Cal/2009 at a given viral titer is therefore more likely to transmit to a contact than an individual infected with Hong Kong/1968 at the same viral titer. A second reason why Cal/2009 has a higher transmission potential than Hong Kong/1968 is because viral titers are higher in Cal/2009-infected contact animals than in Hong Kong/1968-infected contact animals (Fig 1). As such, even if the values of parameter $s$ were the same across the viral isolates, we would expect Cal/2009 to exert a stronger force of infection on susceptible hosts, given that "natural" Cal/2009 infections appear to reach higher viral titers than "natural" Hong Kong/1968 infections. Mutations that either yield increases in viral titers in Hong Kong/1968-infected individuals or those that increase the transmissibility of this virus (per unit viral titer) could both close the gap in transmission

potential between these viruses. Because our approach controls for viral titers when estimating $s$, it could help to disentangle the relative importance of these factors in contributing to pandemic potential. An interesting application of this would be to consider viral strains differing by a handful of mutations, for example an early pandemic strain and one that carries some adaptive mutations. With data from these types of experimental transmission studies, one could identify whether the observed mutations increased transmissibility (controlling for viral titers) or simply increased viral titers that would also impact downstream transmission.

We next used our parameterized force-of-infection functions to estimate $R_0$, an epidemiological parameter commonly used to assess pandemic risk. Because there are many factors relevant to transmission that are not captured by the setup of experimental transmission studies, our estimates of $R_0$ are better interpreted as relative values, rather than absolute values. We therefore conducted all subsequent analyses (Fig 5) across a range of contact rates to account for this uncertainty.

There are several limitations associated with the work presented here. Many of these limitations are intrinsic to using experimental transmission studies for pandemic risk assessment more generally. Most obviously, ferrets are not humans. Results based on transmission experiments in ferrets are only relevant to assessing pandemic risk for humans if transmission potential in ferrets reflects transmission potential in humans. One way to circumvent this is to analyze human experimental transmission studies [16]. However, the number of transmissions will be severely limited in the case of human studies, such that it is unlikely that transmission parameters could be estimated with confidence. Additionally, testing transmission of emerging viruses in humans is unethical because of possible adverse events and side effects. As such, experimental transmission studies in model animals such as ferrets remain the most promising for assessing pandemic risk in humans. Despite the limitations associated with ferrets as a model system, previous analyses do indicate that human infection risk is positively associated with secondary attack rates in ferrets [4], mitigating some of the concern that transmission potential in ferrets does not correlate with transmission potential in humans. Furthermore, frameworks such as the CDC Influenza Risk Assessment Tool [17] rely on experimental transmission studies in animals, highlighting the importance and reliability of these studies for pandemic risk assessment.

Experimental transmission studies, and our framework for analyzing them more quantitatively, are additionally limited by sampling limitations. In transmission studies like those analyzed here, nasal washes or swabs are often taken from a ferret every or every other day. Titers between these measured time points are inferred through interpolation between the available data points. The viral titers we use to estimate the parameters of the force-of-infection functions therefore do not fully capture the entirety of viral population dynamics in infected hosts. For example, it is unlikely that the time of peak viral titer in an animal coincides with the time that a sample is taken from that animal. Our method is therefore likely to systematically overestimate the parameter $s$, as our interpolations miss the viral peaks. Sampling intervals and assay limits of detection also affect when we can say that transmission from index to contact has occurred. If sampling was performed more frequently or assay limits of detection were lower, times of transmission would be more finely resolved. Finally, our inference approach currently assumes that the contact is still uninfected on the last negative test day ($T_1$) prior to the first positive test day ($T_2$). However, this does not necessarily need to be the case: a contact could already be infected at $T_1$ but with viral titers that fall below the limit of detection. Lower limits of detection would thus improve our inference, as would extensions of our approach that would allow for the possibility of infection prior to infection detection.

Another current limitation of our approach is that it assumes that the force-of-infection experienced by a contact animal is related to instantaneous viral titer levels in an index animal. This is necessarily a simplification. It could be that the force of infection at time $t$ depends on viral titers at time $t$ as well as those prior to time $t$. This may happen, for instance, if it takes time for virions to get packaged into droplets prior to these droplets being expelled from the donor animal [18]. We further use viral titer levels as measured by nasal wash. If transmission potential depended more strongly on viral titers at other anatomical sites, our results of relative pandemic risk could be biased. However, several studies using genetically modified viruses have shown that H1N1 and H3N2 viruses transmitted through the air predominately originate from the upper respiratory tract, particularly the nasal respiratory epithelium [19]. Thus, we feel confident that the titer

measurements we used to estimate the transmission function are taken from biologically relevant locations in the ferret respiratory tract. Finally, our model in principle could be extended to consider the impact of symptoms on transmissibility, for example, using previously proposed functional forms [20].

The experimental transmission studies we analyzed used ferrets that were naïve to influenza infection. Clearly, cross-immunity generated from previous infection with seasonally circulating viruses could mitigate the risk of pandemic emergence of certain IAV subtypes. To consider the risk of pandemic emergence in the context of pre-existing immunity, experimental transmission studies could use ferrets with previous immune histories. Indeed, several transmission studies have already been performed using "pre-immune" ferrets [21] and current work of ours is focusing on applying the approach we developed here to data from these studies.

Despite the limitations of our approach and of experimental transmission studies more generally, the method presented here advances integration of pandemic risk assessment with quantitative modeling. We suggest that our approach will be most useful for comparing the pandemic potential of different influenza viruses circulating in non-human reservoirs. Experimental transmission studies are commonly part of an overall risk assessment pipeline that characterizes multiple virological traits thought to be related to pandemic risk [22]. Our data-driven model enriches the insight that can be gained from these experiments. This should help those assessing risk to more accurately weigh pandemic potential and allocate resources appropriately.

## Methods

### Design of the experimental transmission study

Here, we analyze data from a series of experimental transmission studies in ferrets carried out in [6]. The first experimental transmission study used the IAV isolate recombinant A/California/07/2009 (Cal/2009). Index ferrets were inoculated at one of five inoculum doses: $10^0$, $10^1$, $10^2$, $10^4$, and $10^6$ TCID$_{50}$. Four index ferrets were challenged at each of these five doses. Each index ferret was housed in a custom-built transmission cage with a single contact ferret. These cages are designed such that the index and contact ferret are separated by a 5 cm wide double offset perforated divider. This divider allows animals to share the same airspace, but prevents direct physical contact. The contact ferret was introduced one day post index inoculation. Nasal washes were collected every other day from index animals beginning on day 1 post inoculation and from contact animals beginning on day 2 post inoculation (1 day post exposure, dpe). For some experiments, samples were collected up to 11 dpe; for others, samples were collected up to 13 dpe. Virus was titrated using tissue culture infectious dose assays with a limit of detection of $10^1$ TCID$_{50}$/mL. For our analyses, the infection status of an index or contact animal was considered positive if at least one sample had a viral titer measurement that fell above the limit of detection ($10^1$ TCID$_{50}$/mL). The second experimental transmission study used the IAV isolate recombinant A/Hong Kong/1/1968 (Hong Kong/1968). Index ferrets were inoculated at one of six inoculum doses: $10^0$, $10^1$, $10^2$, $10^3$, $10^4$, and $10^6$ TCID$_{50}$. Four index ferrets were challenged at each of these six doses. Other components of the Hong Kong/1968 study were analogous to those of the Cal/2009 study.

### Estimation of the force-of-infection function

We quantified the relationship between viral titer and onward transmission potential using longitudinal viral titer data from paired index-contact ferrets. To do this, we use equations developed to estimate the force of infection using the standard 'catalytic' model [23]. This model assumes that an individual is initially uninfected and susceptible to infection. With time, the individual may become exposed to the pathogen, such that the probability of testing positive increases with time. More quantitatively, the probability that an individual has been infected by time $T$ is given by $1 - e^{-\lambda_c T}$, where $\lambda_c$ is a constant force of infection. This equation has been extended to allow for a time-varying force of infection (e.g., see [24]). In this case, the probability that an individual has been infected by time $T$ is given by $1 - e^{-\int_0^T \lambda(t)dt}$. We adopt this time-varying force-of-infection approach in our analysis of the experimental transmission studies, where the force of infection on the

contact animal comes from the index animal. This force of infection is time-varying because it depends on the viral titers of the index animal and these change over time (see Fig 1).

Adopting the $log_{10}$ functional form, we let the force of infection be given by Equation 1. To evaluate the force of infection $\lambda(t)$ from an index ferret over the course of infection, we therefore need to know that animal's viral titers across all time points. To estimate the complete within-host viral trajectory for each index ferret, we linearly interpolated viral titers on the logarithmic scale. This assumption is equivalent to assuming exponential growth/decay between measured data points. During this fit, we assume that viral titers below the $10^1 TCID_{50}$/mL limit of detection are at $10^{0.5} TCID_{50}$/mL. We further assume that the force of infection $\lambda$ is 0 for all viral titers that fall at or below the limit of detection (LOD) of $10^1 TCID_{50}$/mL.

For each contact ferret, we determine whether the ferret has yet been infected with the focal IAV at each of its measured timepoints. Prior to the first positive viral titer measurement, the ferret is considered uninfected. On the day of the first positive viral titer measurement, the ferret is considered infected. We use these timepoints to estimate the parameter $s$ of the $log_{10}$ force-of-infection function using maximum likelihood. For a contact ferret that does not become infected over the study period, we calculate the probability that this contact remained uninfected by the final study timepoint ($T_f$) for a given value of $s$. This probability is given by:

$$e^{-\int_0^{T_f} \lambda(t)dt} \tag{2}$$

For a contact ferret that does become infected, we compute the probability that this ferret became infected between the time of its last negative test ($T_1$) and the time of its first positive test ($T_2$). This probability is given by the product of the probability that the contact has not become infected by time $T_1$ and the probability that the contact becomes infected between times $T_1$ and $T_2$ times:

$$e^{-\int_0^{T_1} \lambda(t)dt} \times \left(1 - e^{-\int_{T_1}^{T_2} \lambda(t)dt}\right) \tag{3}$$

For any value of $s$, the overall likelihood is given by the product of the likelihoods across all contact ferrets (both those who became infected and those that did not). All but one infected index animal was used to estimate $s$. The one index animal (and corresponding contact animal) we excluded was from the Hong Kong/1968 experiment (at a dose of $10^4$ $TCID_{50}$). We excluded this index-contact pair because the index animal was missing a viral titer measurement at 4 days post exposure.

Given a parameterized force-of-infection function, the probabilities of infection shown in Fig 2B are calculated as:

$$1 - e^{-\lambda_c T} \tag{4}$$

where $T = 1$ day and $\lambda_c$ is the constant force of infection calculated using parameter $s$ and the viral titer given on the x-axis.

To test if estimates of $s$ were significantly different between Cal/2009 and Hong Kong/1968, we estimated a single value for $s$ ("$s_{all}$") based on combined Cal/2009 and Hong Kong/1968 datasets. We then calculated the corrected Akaike information criterion score (AICc) using the $s_{all}$ maximum likelihood estimate. The model with separate $s$ estimates for Cal/2009 and Hong Kong/1968 had a better (smaller) AICc. However, the $\Delta AICc$ scores between this model and the model with only a single $s$ only differed by 2.74 AICc units, indicating that the model with two different $s$ values is not strongly preferred over the model with a single $s$.

## Assessment of alternative force-of-infection functional forms

We considered three alternative functional forms of the relationship between force of infection and viral titers: a linear function, a threshold function, and a Hill function. To parameterize each of these functions, we continue to assume that

viral titers below the LOD do not contribute to the force of infection and thus also do not contribute to the probability of transmission. The linear function assumes that viral titers are linearly related to the force of infection:

$$\lambda(t) = \begin{cases} 0 & \text{if } V(t) < LOD \\ s_L \times V(t) & \text{if } V(t) \geq LOD \end{cases} \tag{5}$$

The threshold function assumes that there is some viral titer threshold $h$ below which the force of infection is zero and above which the force of infection is a constant $r$:

$$\lambda(t) = \begin{cases} 0 & \text{if } V(t) < h \\ r & \text{if } V(t) \geq h \end{cases} \tag{6}$$

The Hill function assumes that viral titers relate to the force of infection according to the following Hill equation:

$$\lambda(t) = \begin{cases} 0 & \text{if } V(t) < LOD \\ q \times \frac{\log_{10}(V)^n}{k_a^n + \log_{10}(V)^n} & \text{if } V(t) \geq LOD \end{cases} \tag{7}$$

To statistically compare these functional forms, we use a maximum likelihood approach to estimate the parameters of each function for both Cal/2009 and Hong Kong/1968. We then use the maximum likelihood values and the number of degrees of freedom of each model to calculate each model's AICc. Table 1 shows these results. Note that for Hong Kong/1968's Hill function estimation, we found that above a $k_a$ value of 300, likelihood values continued to increase until we reached the highest value we considered (a value of $3.8 \times 10^{308}$). However, the increase in likelihood was very slight (<0.01 log-likelihood units over this range). We therefore fixed $k_a$ to 6000 during the inference process and estimated only $q$ and $n$ for Hong Kong/1968's Hill function.

## Quantification of parameters at the transmission event

To project the number of Cal/2009 secondary cases, we used the viral titer dynamics of the Cal/2009 infected contact animals, which we considered our population of "theoretical" index Cal/2009 animals. As we had done previously for the index animals when estimating the parameter $s$, we used linear interpolation to obtain continuous viral titers for each of these contact animals. We performed 1000 stochastic simulations for the Cal/2009 virus. For each stochastic realization, we assumed each animal has an average of $l$ one-hour long contacts per day for the duration of their infection. (In Fig 4, we used $l = 15$ based upon previous estimates for the number of daily contacts for children [11]. We explore the impact of assuming different contact rates in Fig 5.) To simulate daily contacts for each animal, we randomly choose 150 time-points (15 contacts per day × 10 days of infection) and determine the viral titer in the animal at these timepoints. To incorporate uncertainty in $s$, we randomly draw $s$ values from a uniform distribution between 0.001 to 0.2 and accept them with a probability proportional to their likelihood (shown in Fig 2A) using the Metropolis-Hastings algorithm. We draw $s$ values until there are 1000 accepted values and set $s$ to one of these values in each of the 1000 stochastic simulations. With $V(t_{i,d})$ denoting the animal's viral titer for contact number $i$ on day $d$, we then calculate the probability of successful transmission at each of these timepoints using equation 4, with $T = 1$ hour. Given this probability, we then stochastically determine whether a successful transmission occurred at this timepoint. The total number of secondary cases from a given contact animal is then given by summing up the number of successful transmissions across all contacts spanning across all infection days. For each successful transmission, we further store the infection time $t_{i,d}$. This infection time corresponds to the generation interval for this transmission event, defined as the time between infection of an individual and an onward

transmission. In Fig 4C, we plot the distribution of these generation intervals for all onward transmissions in the population of Cal/2009 infected contact animals.

We summarize across the 1000 stochastic realizations as follows. We plot in Fig 4E the mean number of secondary infections from the population of Cal/2009 infected contact animals. This mean number corresponds to the basic reproduction number $R_0$, and we plot the distribution of these $R_0$ values. To estimate the extent of transmission heterogeneity across these realizations, we follow the approach outlined in [13] and estimate the overdispersion parameter $k$ of the negative binomial distribution for each stochastic realization using its simulated distribution of secondary cases. The overdispersion parameter is estimated using the fitdistrplus package in R [25]. We do not estimate $k$ for the subset of stochastic realizations that do not result in any secondary cases from any of the animals. In Fig 4F we plot the cumulative distribution function of this overdispersion parameter for the subset of Cal/2009 stochastic realizations that had at least one onward transmission. Finally, we fit a gamma distribution to the simulated generation intervals from all Cal/2009 stochastic realizations that had at least one onward transmission. We plot the mean ($\frac{\alpha_\gamma}{\lambda_\gamma}$) and coefficient of variation ($\frac{1}{\sqrt{\alpha_\gamma}}$) for each of those simulations in Fig 4G.

Stochastic realizations and summaries of these realizations for Hong Kong/1968 virus were performed in an analogous manner to those for Cal/2009. We statistically compare the $R_0$ estimates for Cal/2009 and Hong Kong/1968 using Welch's two sample t-test.

### Prediction of pandemic potential and dynamics

We estimate $R_0$ across a range of contact rates according to the methods described in the previous section. In Fig 5B, we show the probability of viral establishment across a range of contact rates for two different transmission heterogeneity parameterizations: one in which the overdispersion parameter $k = \infty$ (corresponding to a Poisson distribution of secondary infections across individuals) and one in which the overdispersion parameter $k = 1$ (corresponding to a geometric distribution of secondary infections across individuals). Probabilities of establishment are calculated using the probability generating function:

$$g(s) = (1 + \frac{R_0}{k}(1 - s))^{-k} \tag{8}$$

as described in detail in [13].

In Fig 5C, we plot the average length of a stuttering transmission chain. For contact rates that result in an $R_0 < 1$ for a virus, the average length $\mu$ of a stuttering chain is given by $\mu = \frac{1}{1-R_0}$ [26].

In Fig 5D, we plot the intrinsic growth rate $r$ of a viral epidemic that would establish under a given contact rate. We estimated the intrinsic growth rate assuming a generation intervals that is gamma distributed, such that the growth rate $r$ is given by: $\lambda_\gamma \times (\sqrt[\alpha_\gamma]{R_0} - 1)$ [27]. We identified a representative gamma parameterization where $\alpha_\gamma$=2.18 and $\lambda_\gamma$=0.65 from the results shown in Fig 4G. This corresponds to a mean of 3.34 days with a coefficient of variation of 0.68.

### Supporting information

**S1 Text**.
(PDF)

### Author contributions

**Conceptualization:** Elizabeth Somsen, Anice C. Lowen, Troy C. Sutton, Katia Koelle.

**Data curation:** Kayla M. Septer, Cassandra J. Field, Devanshi R. Patel.

**Formal analysis:** Elizabeth Somsen, Katia Koelle.

**Funding acquisition:** Elizabeth Somsen, Troy C. Sutton, Katia Koelle.

**Investigation:** Elizabeth Somsen.

**Methodology:** Elizabeth Somsen, Katia Koelle.

**Software:** Elizabeth Somsen.

**Supervision:** Katia Koelle.

**Visualization:** Elizabeth Somsen.

**Writing – original draft:** Elizabeth Somsen, Katia Koelle.

**Writing – review & editing:** Elizabeth Somsen, Kayla M. Septer, Cassandra J. Field, Devanshi R. Patel, Anice C. Lowen, Troy C. Sutton, Katia Koelle.

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
