## [Decision Letter · Decision Letter 0]

16 Sep 2025

PCOMPBIOL-D-25-01621

Quantifying viral pandemic potential from experimental transmission studies

PLOS Computational Biology

Dear Dr. Somsen,

Thank you for submitting your manuscript to PLOS Computational Biology. After careful consideration, we feel that it has merit but does not fully meet PLOS Computational Biology's publication criteria as it currently stands. Therefore, we invite you to submit a revised version of the manuscript that addresses the points raised during the review process.

Please submit your revised manuscript within 60 days Nov 16 2025 11:59PM. If you will need more time than this to complete your revisions, please reply to this message or contact the journal office at ploscompbiol@plos.org. Please include the following items when submitting your revised manuscript:

We look forward to receiving your revised manuscript.

Kind regards,

James M McCaw, PhD

Academic Editor

PLOS Computational Biology

Roger Kouyos

Section Editor

PLOS Computational Biology

**Journal Requirements:**

3) Please amend your detailed Financial Disclosure statement. This is published with the article. It must therefore be completed in full sentences and contain the exact wording you wish to be published.

1) If the funders had no role in your study, please state: "The funders had no role in study design, data collection and analysis, decision to publish, or preparation of the manuscript."

2) If any authors received a salary from any of your funders, please state which authors and which funders.

4) Please ensure that the funders and grant numbers match between the Financial Disclosure field and the Funding Information tab in your submission form. Note that the funders must be provided in the same order in both places as well. Currently, the order of this grant (T32AI138952) is different in both places. In addition, "Emory University and the Infectious Diseases Across Scales Training Program (IDASTP, E.D.S.) is missing from the Funding Information tab.

**Reviewers' comments:**

Reviewer's Responses to Questions

Reviewer #1: This study proposes a framework for assessing the transmissibility and pandemic potential of viruses using data from animal transmission experiments. First, the viral load and transmission times from these experiments are used to estimate the transmissibility and overall transmission probability for each virus. Then, contacts in a population are simulated to estimate the basic reproduction number for each virus, accounting for overdispersion of viral loads. Finally, by assuming a fixed generation time, the growth rate is estimated for each pathogen. The modelling framework is sound and can be applied widely to animal experiments conducted to estimate pandemic potential, yielding insights beyond counting the proportion of infected animals as is current practice. The code is well documented and runs on the first try.

Comments:

1. It is unclear why different theoretical generation time distributions have been used to calculate the growth rate, rather than the simulated generation time distributions in Figs 4C-4D (averaged over more realisations).

2. Lines 175-177 “Indeed, a qualitative re-inspection of Figure 1 indicates that, for Cal/2009, neither the peak viral nor the duration of infection in contact animals appears to depend on the index’s inoculum dose” (and similarly for the other virus) – please conduct a formal statistical analysis as this is an open question in the field.

3. The use of “transmissibility” to refer to the parameter s could be reconsidered. “Transmissibility” commonly refers to the parameter beta in population-level models, and so I would expect this to capture R_0 and the transmission probability, rather than only referring to the transmissibility per unit area and then needing to combine “transmissibility” with viral load to get something proportional to R_0. However, I recognise that this recommendation is a personal preference.

4. “In vitro” should be italicised rather than underlined on line 6. Also “a priori” on line 153.

5. Lines 347-351: “Mutations which increased viral titers in Hong Kong/1968 infected individuals to the level of Cal/2009 would close the gap in transmission potential between these viruses. In contrast, if the primary reason for Cal/2009 having higher transmission potential than Hong Kong/1968 was due to higher transmissibility (higher s), then one might expect that mutations that result in higher transmissibility might be selected for in zoonotic Hong Kong/1968 viral populations.” Just because the current gap in R_0 between the viruses is due to differences in viral load, it doesn’t mean that this gap could not be closed by mutations that increase s. It is plausible for viruses to reach the same overall R_0 where one has a higher s and lower viral load, and one has a lower s and higher viral load. To make the original argument, one would have to show that the increase in s required is more difficult to attain evolutionarily than the increase in viral load, and it is unclear how to show that.

6. Consider citing Nishiura et al. (2013) PLOS ONE https://doi.org/10.1371/journal.pone.0055358 for limitations of existing analyses of ferret transmission data.

Reviewer #2: The authors have expanded on an experimental study of influenza A transmission in ferrets. The extension includes data fitting and mathematical modelling to determine the pandemic potential of two different strains.

Comments:

The authors use 'transmission' to include expelling of virus into the environment, exposure to the virus, and then infection of the secondary subject. Transmission can still occur without infection of the secondary. A clear definition of transmission is needed in this study to understand the method, results and conclusion/discussion.

As well, a clear definition of infection is needed.

There is some discussion of the the different viral infection curves between the two strains of the virus. I would like to see some discussion of transmission and infection related to these curves related to high or low viral shedding and large or small sized quanta that the second ferret could be exposed to. Lots of exposure can mean higher viral load at exposure (or even additive exposure over the exposure time window) which can lead to higher probabilities of infection.

Line 111: "As such, viral titer levels appear to impact not only whether a contact animal gets infected but, in the case of infection, when it gets infected." - please revise this sentence and align 'infection' with the definitions of 'transmission' and 'infection' and some discussion of viral load and quanta.

Line 195 - fix the quotation signs that are in the wrong direction

The authors have chosen the log10 transmission function. It would be nice to see that other functions in the supplement i.e., related Figures 4 and 5

Paragraph - lines 306-383 - this paragraph is difficult to read. Please revise for flow and length. It could be much shorter and more succinct.

**Have the authors made all data and (if applicable) computational code underlying the findings in their manuscript fully available?**

Reviewer #1: Yes

Reviewer #2: Yes

PLOS authors have the option to publish the peer review history of their article (what does this mean?). If published, this will include your full peer review and any attached files.

Reviewer #1: **Yes: **Ada Yan

Reviewer #2: No

**Figure resubmission:**
---

## [Decision Letter · Decision Letter 1]

2 Dec 2025

Dear Somsen,

We are pleased to inform you that your manuscript 'Quantifying viral pandemic potential from experimental transmission studies' has been provisionally accepted for publication in PLOS Computational Biology.

Best regards,

James M McCaw, PhD

Academic Editor

PLOS Computational Biology

Roger Kouyos

Section Editor

PLOS Computational Biology

Reviewer's Responses to Questions

**Comments to the Authors:**

Reviewer #1: The authors have sufficiently addressed all points in the initial review.

Reviewer #2: .

**Have the authors made all data and (if applicable) computational code underlying the findings in their manuscript fully available?**

Reviewer #1: Yes

Reviewer #2: Yes

PLOS authors have the option to publish the peer review history of their article (what does this mean?). If published, this will include your full peer review and any attached files.

Reviewer #1: **Yes: **Ada Yan

Reviewer #2: No

---

## [Editor Report · Acceptance letter]

PCOMPBIOL-D-25-01621R1

Quantifying viral pandemic potential from experimental transmission studies

Dear Dr Somsen,

I am pleased to inform you that your manuscript has been formally accepted for publication in PLOS Computational Biology. Your manuscript is now with our production department and you will be notified of the publication date in due course.

With kind regards,

Anita Estes
